# Physical frailty, genetic risk, mediating biomarkers, and risk of suicide attempt: A prospective cohort study

Wei Hu[☯], Li-Jie Gao[☯], Tian-Shu Liu, Ge Tian, Jia-Ning Wang, Yu-Bin Ma, Zi-Ang Zheng, Tong-Jie Feng, Xiao-Xin Niu, Yi-Ning Yan, Bao-Peng Liu*, Cun-Xian Jia*

Department of Epidemiology, School of Public Health, Cheeloo College of Medicine, Shandong University, Jinan, China

☯ These authors contributed equally to this work.
* baopeng.liu@sdu.edu.cn (B-PL); jiacunxian@sdu.edu.cn (C-XJ)

## Abstract

### Background

While physical frailty is linked to psychiatric disorders, its association with suicide attempt (SA) risk is unclear. We aimed to investigate the prospective association of physical frailty with SA risk and the modifying and potential mediating roles of genetic risk and blood biomarkers.

### Methods and findings

This cohort study included 442,920 UK Biobank participants free of SA at baseline. SA events were extracted by linking hospital inpatient records. Physical frailty status was assessed using the five-component Fried phenotype and categorized as nonfrail, prefrail, or frail. Genetic risk for SA was estimated through polygenic risk scores and categorized into high, intermediate, and low risk levels. Cox proportional hazard models were conducted to calculate the hazard ratios (HRs) and 95% confidence intervals (CIs) for the association. Mendelian randomization (MR) analyses were utilized to examine the association between genetically determined physical frailty and SA. Mediation analyses were performed to explore potential biological pathways involving circulating biomarkers. During a median follow-up of 13.6 years, 1,518 (0.3%) individuals developed SA. After multivariable adjustment for sociodemographic characteristics, genetic risk, lifestyle factors, psychiatric disorders, cardiovascular diseases, and cancer, the HRs for SA among those with pre-frailty were 1.61 (95% CI [1.44, 1.80]; $P < 0.001$) and frailty were 2.16 (95% CI [1.78, 2.61]; $P < 0.001$) compared with nonfrail individuals. Genetically predicted frailty was also positively associated with SA (odds ratio = 2.06, 95% CI [1.21, 3.52]; $P = 0.008$). Except for low physical activity, all frailty components were significantly associated with an increased risk of SA (all

**Data availability statement:** Data are available in a public, open access repository. This research has been conducted using the UK Biobank Resource under Application Number 91536. The UK Biobank data are available on application to the UK Biobank (www.ukbiobank.ac.uk/) with access fees.

**Funding:** This work was supported by the Natural Science Foundation of Shandong Province [No: ZR2021QH310 to B.P.L.] https://kjt.shandong.gov.cn; Young Scholars Program of Shandong University [B.P.L.] https://www.sdu.edu.cn; and the National Natural Science Foundation of China [No: 82473710 to C.X.J.] https://www.nsfc.gov.cn. The funders had no role in study design, data collection and analysis, decision to publish, or preparation of the manuscript.

**Competing interests:** The authors have declared that no competing interests exist.

**Abbreviations:** ANOVA, analysis of variance; AP, attributable proportion; CCI, Charlson Comorbidity Index; CIs, 95% confidence intervals; CVD, cardiovascular diseases; DAG, directed acyclic graph; FDR, false discovery rate; FI, frailty index; FP, frailty phenotype; GGT, glutamyltransferase; GWAS, genome-wide association studies; HRs, hazard ratios; HWE, Hardy–Weinberg equilibrium; ICD-10, International Classification of Diseases, 10th Revision; IQR, interquartile range; IVs, instrumental variables; IVW, inverse-variance weighted; KM, Kaplan–Meier; LD, linkage disequilibrium; LOO, leave-one-out; MAF, minor allele frequency; MREC, Multi-center Research Ethics Committee; MR, Mendelian randomization; NHANES, National Health and Nutrition Examination Survey; ORs, odds ratios; PA, physical activity; PARP, population attributable risk percent; PGC, Psychiatric Genomics Consortium; PM, proportion mediated; PRS, polygenic risk scores; RBC, red blood cell; RCS, restricted cubic spline; RERI, relative excess risk due to interaction; SA, suicide attempt; SD, standard deviation; SI, suicidal ideation; SNPs, single nucleotide polymorphisms; STROBE, Strengthening the Reporting of Observational Studies in Epidemiology; TBIL, total bilirubin; TDI, Townsend Deprivation Index; TP, total protein; WBC, white blood cell.

$P < 0.05$), with HRs ranging from 1.16 (95% CI [1.01, 1.33]; $P = 0.038$) to 1.62 (95% CI [1.43, 1.82]; $P < 0.001$). The additive interaction of physical frailty and genetic risk increased the risk of SA, with the highest risk observed among frail individuals with high genetic risk (HR = 3.09, 95% CI [2.30, 4.17]; $P < 0.001$), whereas no significant multiplicative interactions were detected. Biomarkers related to liver function, metabolism, immunity, and inflammation may have partially explained this association, accounting for a collective 14.15% (95% CI: 8.94%, 19.74%; $P < 0.001$) of the total effect, although MR analyses did not support causal mediating effects. The key limitations of this analysis include potential residual confounders and the limited representativeness of the study population.

### Conclusions

Pre-frail and frail states were associated with an increased risk of SA, especially among individuals with high genetic risk. Incorporating frailty assessment and management into primary prevention strategies may have implications for SA prevention.

---

## Author summary
### Why was this study done?

- Suicide attempt (SA) is a major public health concern, and identifying risk factors may help improve early prevention.

- Physical frailty has been linked to poor mental health, but whether it is related to the risk of SA is not well understood.

- This study aimed to examine whether physical frailty is associated with the risk of SA and whether genetic risk and blood biomarkers influence this association.

### What did the researchers do and find?

- We analyzed data from more than 440,000 adults without a history of SA at baseline in a large population-based cohort.

- Frail individuals had a substantially higher risk of SA compared with nonfrail individuals, and genetic analyses provided additional support for this association.

- Participants who were both physically frail and at high genetic risk had the highest risk of SA, and several blood markers related to metabolism, liver function, inflammation, and immune response may partly explain this relationship.

### What do these findings mean?

- Physical frailty may represent an important and previously under-recognized risk factor for SA, particularly among individuals with a higher genetic risk for SA.

- Assessing and managing frailty could help detect individuals at higher risk and inform suicide prevention strategies.

- Because this study was observational and based mainly on participants of European ancestry in the UK Biobank, the findings cannot prove cause and effect and may not be generalizable to other populations.

## Introduction

Suicidal behaviors, encompassing suicide attempt (SA) and suicide death, represent a major and often underestimated global public health challenge [1]. In 2021, an estimated 746,000 people died by suicide worldwide [2]. For every suicide death, there are more than 20 SA [3]. SA, one of the strongest risk factors for suicide deaths, may be preventable [1]. Accordingly, identifying modifiable risk factors for SA is essential for reducing suicide mortality. Although existing research on SA has largely focused on psychiatric disorders and psychosocial determinants, emerging evidence has suggested that aging-related vulnerability may play an important role in suicide risk [1,3]. Suicide death rates increase with age [4], and SA among older adults is more likely to be fatal [3], suggesting that age-related declines in physiological resilience and stress tolerance may contribute to greater vulnerability to suicidal behavior.

Physical frailty, representing a modifiable aging process, is characterized by reduced multisystem physiological reserve and heightened vulnerability to stressors, leading to impaired homeostasis and increased risk of adverse outcomes such as disability, hospitalization, and mortality [5,6]. Frailty has been increasingly recognized as relevant to mental health outcomes [7–9]. Conceptually, frailty may heighten susceptibility to suicide-related outcomes through multiple interrelated pathways, including functional impairment, loss of independence, chronic inflammation, endocrine dysregulation, metabolic disturbances, and a reduced capacity to cope with psychological and physical stressors [5,6,10–15]. In line with this perspective, a recent narrative review has highlighted frailty as a potentially important factor in late-life suicidal ideation (SI) and SA, although the existing evidence is largely qualitative in nature [16]. Despite this biological and clinical plausibility, empirical evidence linking frailty to suicide-related outcomes remains limited. Existing quantitative studies on this topic have predominantly focused on SI, consistently reporting positive associations between frailty and SI [17–19]. However, SA is more strongly associated with suicide mortality than SI, and evidence specifically addressing SA is scarce. To date, only one quantitative study has examined the association between frailty and SA, using a cumulative-deficit frailty index (FI) [15]. While informative, the FI is complex and may be less feasible for routine clinical or large-scale epidemiological use. Unlike the complex FI, the frailty phenotype (FP), which captures frailty based on five components [fatigue, weakness, slowness, weight loss, and low physical activity (PA)], is a widely adopted, clinically intuitive, and modifiable measure of physical frailty [5,20]. However, the association between FP-measured physical frailty and SA remains unexplored. Importantly, previous studies have identified individual components of the FP associated with suicide-related outcomes, including fatigue [21], reduced grip strength [17,22], slow walking speed [17], weight loss [23], and low PA [24]. These observations support a biologically and clinically plausible hypothesis that FP-measured physical frailty may be associated with increased risk of SA. Hence, a comprehensive study integrating a prospective cohort design with Mendelian randomization (MR) is warranted to better characterize this association.

Beyond behavioral and functional mechanisms, genetic and biological factors may further shape the frailty-SA relationship. Genome-wide association studies (GWAS) have highlighted the role of genetic factors in the development of SA and identified numerous single-nucleotide polymorphisms (SNPs) associated with SA [25]. Despite the limited predictive power of individual SNPs, the polygenic risk scores (PRS), which quantifies genetic risk by considering the combined effects of multiple SNPs, have shown robust associations with SA [13]. At the same time, both frailty and SA have been linked to systemic inflammation, endocrine dysfunction, and metabolic dysregulation [5,6,10–15]. Understanding whether genetic risk to SA modifies the impact of frailty, and whether shared biological pathways mediate this association, may provide insight into underlying mechanisms and inform targeted prevention strategies.

Against this background, the present study aimed to examine the association between FP-measured physical frailty and the risk of SA in a large prospective cohort from the UK Biobank. We further employed two-sample MR analyses, as a complementary approach, to assess whether genetically predicted frailty shows evidence consistent with the observed association. Additionally, we evaluated whether genetic risk for SA modifies this association and explored the potential mediating role of a range of blood biomarkers. We hypothesized that physical frailty would be associated with an increased risk of SA, that this association would be stronger among individuals with higher genetic risk, and that biological pathways reflected by blood biomarkers may partially mediate this relationship.

## Methods

### Ethics statement and consent to participate

The Northwest Multi-center Research Ethics Committee (MREC reference: 21/NW/0157) provided ethical approval for the UK Biobank project. All participants gave written informed consent before being recruited.

### Study design and participants

The study population was derived from the UK Biobank, a large-scale prospective cohort of over half a million middle-aged and older participants, recruited from 22 assessment centers across England, Scotland, and Wales between 2006 and 2010. The cohort's details have been well-documented in a previous publication [26]. Data were gathered from participants who gave written informed consent before recruitment via touch-screen questionnaires, interviews, and physical examinations.

Out of 502,301 available participants' data, we excluded those who were withdrawn or lost to follow-up ($n = 1,297$) and those diagnosed with a history of SA at baseline ($n = 2,789$). Next, participants with incomplete information on frailty components ($n = 37,211$) were excluded. To define genetically informed ancestry for analyses involving PRS, we used genetically inferred ancestry assignments following the Pan-UK Biobank (Pan-UKBB) framework [27], which compares each participant's genome-wide genotype data to global reference panels (HGDP and the 1000 Genomes Project) and assigns individuals to the ancestry group they are most genetically similar to, while excluding individuals without a confident assignment. This assignment is based on genetic similarity and does not rely on self-reported ethnicity [28]. Because the genetic tool used to compute the PRS for SA was derived and validated primarily in individuals of European genetic ancestry [25], we excluded participants with missing genetic data or assigned to nonEuropean genetically inferred ancestry groups ($n = 11,253$). Finally, participants with incomplete information on covariates were excluded ($n = 6,831$), leaving 442,920 eligible participants for the association analysis (Fig A in S1 Appendix). The current study was reported according to the Strengthening the Reporting of Observational Studies in Epidemiology (STROBE) guideline (S1 Checklist). This study was conducted to test a priori hypotheses using data from the UK Biobank. A prospective analysis plan was not formally published prior to the analyses. The primary analytical framework was specified in advance and was conducted between March 2025 and May 2025. No data-driven changes were made to the primary analyses after inspection of the main results.

### Assessment of physical frailty

We utilized the FP developed by Fried [29] and later revised by Hanlon and colleagues [20] to accommodate data from the UK Biobank and thereby quantify the physical frailty status of study participants. The FP consists of five components: weight loss, exhaustion, low PA, slow gait speed, and low grip strength [20,29]. Each component was scored 1 if the specified criteria were met and 0 otherwise, resulting in a total score ranging from 0 to 5, with higher scores reflecting greater frailty [20,29]. The definitions and criteria for each component are outlined in Table A in S2 Appendix. Following previous studies [20,29], participants were categorized as nonfrailty (0 points), pre-frailty (1–2 points), or frailty (≥3 points).

## Assessment of SA

Hospital inpatient records (Data-Fields 41,270 and 41,271), including admission and diagnosis data, were used to identify SA cases. These records were sourced from the Hospital Episode Statistics for England, the Scottish Morbidity Record for Scotland, and the Patient Episode Database for Wales [26]. Incident SA cases during follow-up were ascertained exclusively from hospital inpatient records, based on diagnoses recorded under the International Classification of Diseases, 10th Revision (ICD-10) coding system (X60-X84, Y870) [30]. Self-reported information was not used to define SA events during follow-up. The follow-up time was computed as the interval from baseline recruitment to the date of first diagnosis, death, or the censoring date (31 October 2022 for England, 31 August 2022 for Scotland, and 31 May 2022 for Wales), whichever occurred first. Baseline SA cases were identified using both hospital inpatient records and self-reported information collected at baseline (Data-Field 20002, code 1290). Participants with evidence of SA from either source prior to baseline were excluded from the analysis to ensure that all SA events analyzed in the study represented incident cases occurring after baseline.

## Genetic risk for SA

This study utilized genetic data from the UK Biobank, restricting analyses to participants of European genetic ancestry, selected following established procedures [31]. Detailed descriptions of the genotyping and quality control procedures implemented by the UK Biobank have been reported previously [31]. Variants were filtered using standard quality control criteria, including minor allele frequency (MAF) ≥ 0.01, imputation INFO score ≥ 0.3, Hardy-Weinberg equilibrium (HWE) $P$-value ≥ $1 \times 10^{-6}$, and a genotyping rate of ≥95%. After filtering, 658,886 variants remained for PRS construction. The PRS for SA were derived using GWAS summary statistics from a large meta-analysis of SA in individuals of European genetic ancestry [25]. The originally published Psychiatric Genomics Consortium (PGC) GWAS meta-analysis included 15 cohorts, one of which was the UK Biobank. To avoid sample overlap between the discovery GWAS and the UK Biobank target sample used in the present study, and consistent with previous studies [13], we utilized a re-generated version of the GWAS summary statistics in which the UK Biobank cohort was explicitly excluded. This resulted in a new GWAS meta-analysis based on the remaining 14 independent cohorts, comprising 33,353 SA cases and 444,626 controls. The GWAS summary statistics used in the present study were obtained directly from the PGC through the Suicide Attempt Data Access Portal (https://pgc.unc.edu/for-researchers/data-access-committee/data-access-portal/) [accessed: 28/04/2025]. The PRS for SA were constructed using PRS-CS [32], a Bayesian polygenic prediction method that estimates SNP effect sizes by integrating GWAS summary statistics with external linkage disequilibrium (LD) reference panels. Given that both the discovery GWAS and the target dataset consisted of individuals of European genetic ancestry, LD was estimated using the European reference panel from the 1000 Genomes Project [33]. All eligible SNPs were combined to calculate the PRS for each participant using PLINK software. Higher PRS values were associated with increased genetic risk for SA. We standardized the PRS and classified participants into low, intermediate, and high genetic risk groups according to tertiles.

## Assessment of biomarkers

Consenting participants provided blood samples at baseline, which were separated into components and stored at the UK Biobank (−80 °C and LN2). Blood biomarkers, which underwent rigorous quality control, have been externally validated [34]. Guided by prior knowledge and existing literature [5,6,8,10–15], 61 blood biomarkers reflecting diverse biological pathways, such as liver and kidney function, immunometabolism, endocrine function, red and white blood cells, bone health, and platelets, were selected as potential mediators linking physical frailty and SA. Further details on the selected biomarkers are provided in the S1 Text and Table B in S2 Appendix.

## Covariates

Sociodemographic factors [age (continuous), sex, educational level (college or university versus others), employment status (employed versus unemployed), Townsend Deprivation Index (TDI, continuous, with higher values indicating greater deprivation)], lifestyle factors [smoking status (previous/never versus current), drinking frequency (≥3 times/week versus less), and body mass index (BMI, <25 versus versus ≥25 kg/m$^2$)], and medical histories [cardiovascular diseases (CVD), psychiatric disorders, and cancer] were included as potential covariates. Psychiatric disorders were defined as the presence of any mental or behavior disorders (ICD-10: F00-F99) from the '*First occurrence fields*' of health-related outcomes in the UK Biobank [35]. These covariates were selected a *priori* based on previous studies demonstrating their associations with both physical frailty and SA [3,6,13], with a directed acyclic graph (DAG) additionally used to support their identification as potential confounders (Fig B in S1 Appendix). Detailed information on the definition of covariates is provided in Table C in S2 Appendix. A summary of missing data is provided in Table D in S2 Appendix, with the highest proportion of missing values recorded for educational level (0.87%).

## Statistical analyses

Descriptive statistics were performed for continuous variables [mean (standard deviation, SD) or median (interquartile range, IQR)] and categorical variables [n (%)] to summarize the baseline characteristics of study samples by frailty or SA status. Differences in these characteristics were compared using *t*-tests or one-way analysis of variance (ANOVA) for normally distributed continuous variables, Kruskal–Wallis rank-sum tests for nonnormally distributed continuous variables, and chi-squared tests for categorical variables.

After confirming that the proportional hazards assumption was not violated through the Schoenfeld residuals test, Cox proportional hazards models with follow-up time as the timescale were performed to calculate hazard ratios (HRs) and 95% confidence intervals (CIs). First, the frailty score was treated as a continuous variable. Second, frailty status was analyzed as a tri-categorical variable, using nonfrail individuals as the reference group. The same procedure was applied to the PRS for SA. Additionally, the population attributable risk percent (PARP) was calculated to assess the contribution of frailty status to SA [36]. Third, associations between individual frailty components and SA risk were examined under mutual adjustment for the other components. The cumulative hazard of SA across frailty statuses was compared using the Kaplan–Meier (KM) method with the log-rank test. Four Cox models with stepwise adjustments for covariates were conducted: Model 0 was an unadjusted model; Model 1 adjusted for sociodemographic factors and genetic risk for SA (or frailty status in the genetic risk model); Model 2 additionally adjusted for lifestyle factors; and Model 3 (the primary model) further incorporated medical histories. The first 10 principal components of ancestry were included as covariates in the genetic-related analyses. A restricted cubic spline (RCS) model adjusted for covariates in Model 3 was constructed to explore the dose-response relationship between frailty score and SA.

Two-sample MR analyses were conducted as a secondary and supportive analysis to provide genetic evidence. Compared with conventional observational analyses, this approach is less susceptible to confounding and reverse causation [37]. To ensure the reliability of MR analysis, the study adhered to three core assumptions: (1) instrumental variables (IVs) must be significantly correlated with the exposure factor (correlation assumption); (2) IVs must be independent of confounding factors (independence assumption); and (3) IVs must influence the outcome variable solely through the target exposure (exclusivity assumption). IVs for physical frailty were obtained from the UK Biobank GWAS summary statistics [38], whereas IVs for SA were obtained from the PGC GWAS used to construct the PRS for SA [25]. A total of 30 SNPs that were highly associated with physical frailty ($P < 5 \times 10^{-8}$) were used as IVs (Table E in S2 Appendix). We estimated odds ratios (ORs) and 95% CIs. Our primary MR analysis employed the inverse-variance weighted (IVW) method. To assess robustness, we conducted sensitivity analyses using the MR-Egger regression and weighted median method. Horizontal pleiotropy was evaluated via the MR-Egger intercept, and heterogeneity was tested with Cochran's Q statistic.

Leave-one-out (LOO) analyses identified whether any single SNP drove the causal estimates. Further methodological details are provided in the S1 Text. The MR study followed the STROBE-MR guidelines (S2 Checklist) [39].

Using participants with nonfrailty and low genetic risk as the reference group, the joint associations between frailty status, genetic risk, and SA risk were investigated. Moreover, the associations between frailty status and SA were estimated stratified by genetic risk. A fully-adjusted Cox model was conducted to assess multiplicative interaction by introducing a product term (frailty status × genetic risk). An additive interaction model was evaluated using the relative excess risk due to interaction (RERI) and the attributable proportion due to interaction (AP). An additive effect was deemed significant if the 95% CIs for both the RERI and AP did not include 0.

Consistent with previous studies [40], we fitted two models to identify potential biomarkers mediating the frailty-SA association. First, after adjusting for covariates in Model 3 and the frailty score, Cox models were constructed to examine associations between selected biomarkers and SA. Second, biomarkers demonstrating statistical significance in the Cox model were included as dependent variables in fully-adjusted linear regression models to assess their associations with the frailty score. Biomarkers that were statistically significant and showed a consistent direction of effect in both models were identified as potential mediators and included in the formal mediation analysis. The proportion mediated (PM) for each biomarker was estimated using the '*mediation*' package in R software, with 95% CIs calculated via a nonparametric bootstrap method of sampling 1,000 times. Additionally, the difference method was applied to evaluate the overall PM of all significant mediators in the frailty-SA association [41]. To further investigate whether the biomarkers identified in the observational mediation analysis lie on the causal pathway linking physical frailty to SA, we conducted two-step MR analyses [42]. In the first step, genetic instruments for physical frailty were used to estimate the causal effect of frailty on each candidate biomarker. In the second step, genetic instruments for the corresponding biomarker were applied to assess its causal effect on SA. Details regarding the selection of genetic instruments, quality control procedures, and MR estimation methods are provided in the S1 Text. Evidence for causal mediating roles was considered present only if both steps demonstrated statistically significant associations in the expected direction.

To assess the robustness of the primary findings, we conducted a series of pre-specified sensitivity analyses addressing potential sources of bias and confounding. First, to mitigate potential reverse causation, we excluded SA cases occurring within the first 2 years of follow-up and re-estimated the associations. Second, to account for missing covariate data, we repeated the association analyses using imputed data, processing missing covariates via multiple imputation techniques. Specifically, we generated five imputed datasets with 20 iterations to ensure convergence and stability of the imputation process and pooled the estimates from Cox proportional hazards regression models across imputed datasets using Rubin's rules [43]. Third, to further minimize potential confounding from severe preexisting physical and mental conditions and socioeconomic disadvantage, additional sensitivity analyses were conducted excluding participants with baseline CVD, cancer, or psychiatric disorders, as well as those with greater socioeconomic deprivation. Fourth, to further address potential residual confounding by comorbidities, we additionally adjusted for the Charlson Comorbidity Index (CCI) (Table F in S2 Appendix) [44]. We also calculated E-values to quantify the minimum strength of association that an unmeasured confounder would need to have with both frailty status and SA to fully explain the observed associations [45]. Fifth, given that death could preclude the occurrence of SA during follow-up, we conducted a competing risk analysis using the Fine-Gray subdistribution hazards model, treating death as a competing event. The detailed results of these sensitivity analyses are presented in the Supplementary Materials. Moreover, the frailty-SA association stratified by sex, age, education level, smoking status, drinking frequency, and BMI was examined.

All analyses were conducted using SAS 9.4 (SAS Institute, Cary, NC, USA) and R software version 4.4.2 (R Foundation for Statistical Computing). Statistical significance was defined as a Benjamin–Hochberg false discovery rate (FDR)-corrected *P*-value ($P_{FDR}$) < 0.05 for analyses involving biomarkers, and a two-sided *P*-value < 0.05 for all other analyses.

## Results

### Baseline characteristics

During a median follow-up of 13.6 years (IQR, 12.9, 14.2 years), 1,518 participants (0.3%) experienced SA. Among the included 442,920 participants [mean age (SD), 56.5 (8.1) years; 54.1% female], 58.3% were classified as nonfrail, 38.2% as pre-frail, and 3.5% as frail. Frail participants were older, more likely to be female, had lower educational attainment, were unemployed, experienced more deprivation, were current smokers, had a higher BMI, and were more likely to suffer from baseline diseases compared to nonfrail participants (Table 1; all $P < 0.001$). Similar patterns were observed when baseline characteristics were stratified by SA status (Table G in S2 Appendix).

### Independent associations of frailty and genetic risk with SA

The KM curve showed that the cumulative risk of SA was significantly higher in frail participants (Fig C in S1 Appendix; log-rank $P < 0.001$). The RCS analysis revealed a nonlinear dose-response relationship between the frailty score and SA risk, characterized by an initial rise followed by a plateau (nonlinear $P = 0.001$, Fig D in S1 Appendix). Each 1-point

**Table 1. Baseline characteristics according to frailty status.**

| Characteristics | Overall | Frailty status | | | P values |
|---|---|---|---|---|---|
| | | Nonfrailty | Pre-frailty | Frailty | |
| | (*N* = 442,920) | (*N* = 258,139) | (*N* = 169,087) | (*N* = 15,694) | |
| Age, mean (SD) | 56.5 (8.1) | 56.3 (8.1) | 56.7 (8.1) | 58.1 (7.6) | <0.001 |
| Sex (female), *n* (%) | 239,686 (54.1) | 134,206 (52.0) | 95,713 (56.6) | 9,767 (62.2) | <0.001 |
| Education level, *n* (%) | | | | | <0.001 |
| College or university degree, *n* (%) | 147,969 (33.4) | 96,164 (37.3) | 49,226 (29.1) | 2,579 (16.4) | |
| Others | 294,951 (66.6) | 161,975 (62.7) | 119,861 (70.9) | 13,115 (83.6) | |
| Employment status, *n* (%) | | | | | <0.001 |
| Employed[a] | 409,567 (92.5) | 244,615 (94.8) | 153,900 (91.0) | 11,052 (70.4) | |
| Unemployed | 33,353 (7.5) | 13,524 (5.2) | 15,187 (9.0) | 4,642 (29.6) | |
| Townsend deprivation index, median (IQR) | −2.2 (−3.7, 0.4) | −2.5 (−3.8, −0.2) | −1.9 (−3.5, 1.0) | 0.1 (−2.6, 3.1) | <0.001 |
| Drinking frequency, *n* (%) | | | | | <0.001 |
| <3 times/week | 245,868 (55.5) | 129,513 (50.2) | 104,111 (61.6) | 12,244 (78.0) | |
| ≥3 times/week | 197,052 (44.5) | 128,626 (49.8) | 64,976 (38.4) | 3,450 (22.0) | |
| Smoking status, *n* (%) | | | | | <0.001 |
| Previous/never smoking | 397,947 (89.8) | 236,060 (91.4) | 149,185 (88.2) | 12,702 (80.9) | |
| Current smoking | 44,973 (10.2) | 22,079 (8.6) | 19,902 (11.8) | 2,992 (19.1) | |
| BMI, *n* (%) | | | | | |
| <25 kg/m$^2$ | 147,907 (33.4) | 99,724 (38.6) | 45,596 (27.0) | 2,587 (16.5) | <0.001 |
| ≥25 kg/m$^2$ | 295,013 (66.6) | 158,415 (61.4) | 123,491 (73.0) | 13,107 (83.5) | |
| Medical histories at baseline, *n* (%) | | | | | |
| Psychiatric disorders (yes) | 73,022 (16.5) | 34,276 (13.3) | 33,520 (19.8) | 5,226 (33.3) | <0.001 |
| CVD (yes) | 28,942 (6.5) | 11,910 (4.6) | 13,767 (8.1) | 3,265 (20.8) | <0.001 |
| Cancer (yes) | 40,078 (9.0) | 22,140 (8.6) | 15,962 (9.4) | 1,976 (12.6) | <0.001 |

*Note*: [a]Employed included those in paid employment or self-employed, retired, doing unpaid or voluntary work, or being full or part time students.

The significance of differences between groups was assessed using one-way analysis of variance for normally distributed continuous variables, the Kruskal–Wallis rank-sum test for nonnormally distributed continuous variables, and the chi-squared test for categorical variables. *P*-values were calculated based on these respective tests.

Abbreviations: BMI, body mass index; CVD, cardiovascular diseases; SD, standard deviation; IQR, interquartile range.

increment in the frailty score was associated with a 29% (HR = 1.29, 95% CI [1.23, 1.36]; *P* < 0.001) higher risk of SA (Table 2). Compared to nonfrail individuals, the fully-adjusted HRs for SA were 1.61 (95% CI [1.44, 1.80]; *P* < 0.001) in pre-frail individuals and 2.16 (95% CI [1.78, 2.61]; *P* < 0.001) in frail individuals (*P* for trend < 0.001) (Table 2). The PARP analysis indicated that 21.3% (95% CI: 15.5%, 27.1%) of SA cases could potentially be prevented if all individuals maintained a nonfrail status (Table 2). In secondary analyses, MR results were directionally consistent with the findings from the cohort analysis, showing that genetically determined frailty was associated with an increased risk of SA (OR = 2.06, 95% CI [1.21, 3.52]; *P* = 0.008) (Fig 1). In PRS analyses, participants with a high genetic risk had a 62% (HR = 1.62, 95% CI [1.42, 1.85]; *P* < 0.001) higher risk of SA compared to those with a low genetic risk (Table 2). Per SD increment in the PRS for SA was associated with a 25% (HR = 1.25, 95% CI [1.19, 1.32]; *P* < 0.001) higher risk of SA (Table 2).

**Table 2. Associations of frailty status and genetic risk with the risk of suicide attempt.**

| | Cases/ person-years | Model 0 | | Model 1 | | Model 2 | | Model 3 | |
|---|---|---|---|---|---|---|---|---|---|
| | | HR (95% CI) | *P* | HR (95% CI) | *P* | HR (95% CI) | *P* | HR (95% CI) | *P* value |
| **Frailty status**[a] | | | | | | | | | |
| Nonfrail (0) | 570/3,454,579 | 1(Reference) | | 1(Reference) | | 1(Reference) | | 1(Reference) | |
| Pre-frail (1–2) | 782/2,218,892 | 2.13 (1.91, 2.38) | <0.001 | 1.86 (1.67, 2.08) | <0.001 | 1.80 (1.61, 2.02) | <0.001 | 1.61 (1.44, 1.80) | <0.001 |
| Frail (3–5) | 166/194,313 | 5.14 (4.32, 6.11) | <0.001 | 3.06 (2.54, 3.69) | <0.001 | 2.79 (2.30, 3.38) | <0.001 | 2.16 (1.78, 2.61) | <0.001 |
| *P* for trend | | <0.001 | | <0.001 | | <0.001 | | <0.001 | |
| Per 1-point increase | 1,518/5,867,785 | 1.65 (1.58, 1.73) | <0.001 | 1.44 (1.37, 1.51) | <0.001 | 1.40 (1.33, 1.47) | <0.001 | 1.29 (1.23, 1.36) | <0.001 |
| PARP[b] | | 36.4% (32.2%, 40.5%) | | 28.5% (23.5%, 33.4%) | | 27.4% (22.3%, 32.6%) | | 21.3% (15.5%, 27.1%) | |
| **Genetic risk**[c] | | | | | | | | | |
| Low | 337/1,958,029 | 1(Reference) | | 1(Reference) | | 1(Reference) | | 1(Reference) | |
| Intermediate | 506/1,956,827 | 1.50 (1.31, 1.73) | <0.001 | 1.42 (1.23, 1.63) | <0.001 | 1.40 (1.21, 1.60) | <0.001 | 1.36 (1.18, 1.56) | <0.001 |
| High | 675/1,952,929 | 2.01 (1.76, 2.29) | <0.001 | 1.77 (1.55, 2.02) | <0.001 | 1.72 (1.50, 1.96) | <0.001 | 1.62 (1.42, 1.85) | <0.001 |
| Per SD increment | 1,518/5,867,785 | 1.38 (1.31, 1.45) | <0.001 | 1.30 (1.24, 1.37) | <0.001 | 1.29 (1.22, 1.36) | <0.001 | 1.25 (1.19, 1.32) | <0.001 |

*Note*: [a]Frailty status was assessed using a frailty phenotype score ranging from 0 to 5 and categorized as nonfrail (0), pre-frail (1–2), and frail (3–5). [b]PARP indicated the proportion of suicide attempts that could theoretically be prevented in this study population if the entire population could have been maintained nonfrail. Population attributable risk at the median follow-up time of the study population was reported. [c]Genetic risk was evaluated using a polygenic risk score and categorized into low, intermediate, and high genetic risk based on tertiles of the score distribution. HRs (95% CIs) and *P*-values were estimated using Cox proportional hazard models. Model 0 was an unadjusted model. For the frailty status model, Model 1 was adjusted for age, sex, education, employment, Townsend Deprivation Index, and genetic risk. For the genetic risk model, Model 1 was adjusted for age, sex, education, employment, Townsend Deprivation Index, frailty status, and the first 10 principal components of ancestry. Model 2 was adjusted for Model 1 plus smoking status, drinking frequency, and body mass index. Model 3 was adjusted for Model 2 plus preexisting psychiatric disorders, cardiovascular diseases, and cancer. Abbreviations: HR, hazards ratio; CI, confidence interval; SD, standard deviation; PARP, population attributable risk percent.

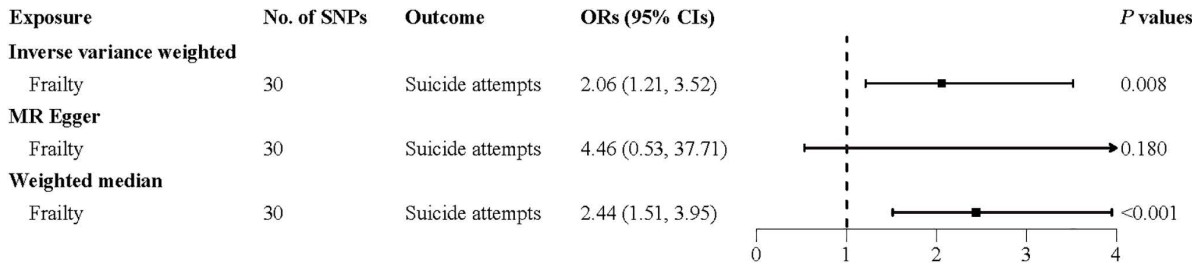

**Fig 1. Two-sample MR analyses for the associations of genetically predicted physical frailty with the risk of suicide attempt.** *Note*: ORs (95% CIs) and *P*-values were estimated using two-sample MR analyses. Abbreviations: SA, suicide attempt; OR, odds ratio; CI, confidence interval; MR, Mendelian randomization; SNP, single nucleotide polymorphisms.

After adjusting for covariates in Model 3, all frailty components were significantly associated with SA risk. The HRs are as follows: 1.42 (95% CI [1.26, 1.60]; *P* < 0.001) for weight loss, 1.69 (95% CI [1.51, 1.91]; *P* < 0.001) for exhaustion, 1.27 (95% CI [1.11, 1.45]; *P* < 0.001) for low PA, 1.43 (95% CI [1.23, 1.66]; *P* < 0.001) for slow gait speed, and 1.26 (95% CI [1.10, 1.44]; *P* < 0.001) for low grip strength (Table 3). When further adjusting the frailty components for each other, we discovered the following HRs for SA risk: 1.42 (95% CI [1.26, 1.60]; *P* < 0.001) for weight loss, 1.62 (95% CI [1.43, 1.82]; *P* < 0.001) for exhaustion, 1.13 (95% CI [0.98, 1.30]; *P* = 0.101) for low PA, 1.20 (95% CI [1.02, 1.41]; *P* = 0.025) for slow gait speed, and 1.16 (95% CI [1.01, 1.33]; *P* = 0.038) for low grip strength (Table 3).

## Joint and interaction analyses of frailty and genetic risk with SA

Compared with nonfrail participants with low genetic risk, frail participants with high genetic risk showed the highest risk of SA (HR = 3.09, 95% CI [2.30, 4.17]; *P* < 0.001) (Fig 2). When using frail participants as the reference group, the risk of SA among nonfrail participants was similar across both genetic risk groups, with reductions of 61% (HR = 0.39, 95% CI [0.26, 0.58]; *P* < 0.001) in the low genetic-risk group, 54% (HR = 0.46, 95% CI [0.33, 0.64]; *P* < 0.001) in the intermediate-genetic-risk group, and 54% (HR = 0.46, 95% CI [0.35, 0.61]; *P* < 0.001) in the high genetic-risk group (*P* for multiplicative interaction = 0.198, Table H in S2 Appendix). The RERI and AP suggested an additive effect of frailty status and genetic risk on SA risk (Table I in S2 Appendix). Specifically, compared to individuals with nonfrailty and low genetic risk, those with pre-frailty/frailty and high genetic risk had a RERI of 0.515 (95% CI: 0.113, 0.917) and an AP of 0.195 (95% CI: 0.045, 0.345).

## Mediation analyses

The associations between selected biomarkers and SA, as well as the relationships between the frailty score and these biomarkers, are presented in Tables J and K in S2 Appendix, respectively. A total of 20 biomarkers were associated with both the frailty score and SA ($P_{FDR}$ < 0.05). However, five biomarkers were excluded due to having the opposite direction of effect in the two models. As shown in Fig 3, 15 biomarkers were ultimately considered potential mediators and included in the mediation analyses. These biomarkers included gamma-glutamyltransferase (GGT), total bilirubin (TBIL), total protein (TP), glucose, red blood cell (RBC) count, hemoglobin concentration, hematocrit percentage, mean reticulocyte volume, mean sphered cell volume, immature reticulocyte fraction, white blood cell (WBC) count, neutrophil count, lymphocyte percentage, neutrophil percentage, and platelet count. Each of these biomarkers demonstrated a statistically significant indirect effect in the observational mediation analyses (all $P_{FDR}$ < 0.05), with the PM ranging from 0.23% (95% CI: 0.06%,

**Table 3. Associations of individual components of frailty with the risk of suicide attempt.**

| Frailty components | Cases/ person-years | Model 0 | | Model 3 | | Mutual adjustment model | |
|---|---|---|---|---|---|---|---|
| | | HR (95% CI) | *P* value | HR (95% CI) | *P* value | HR (95% CI) | *P* value |
| Weight loss | 346/886,717 | 1.66 (1.47, 1.87) | <0.001 | 1.42 (1.26, 1.60) | <0.001 | 1.42 (1.26, 1.60) | <0.001 |
| Exhaustion | 447/701,185 | 3.07 (2.75, 3.43) | <0.001 | 1.69 (1.51, 1.91) | <0.001 | 1.62 (1.43, 1.82) | <0.001 |
| Low physical activity | 280/574,634 | 2.08 (1.83, 2.37) | <0.001 | 1.27 (1.11, 1.45) | <0.001 | 1.13 (0.98, 1.30) | 0.101 |
| Slow walking pace | 250/416,120 | 2.57 (2.24, 2.94) | <0.001 | 1.43 (1.23, 1.66) | <0.001 | 1.20 (1.02, 1.41) | 0.025 |
| Low grip strength | 277/779,732 | 1.45 (1.27, 1.65) | <0.001 | 1.26 (1.10, 1.44) | <0.001 | 1.16 (1.01, 1.33) | 0.038 |

*Note*: Model 0 was an unadjusted model. Model 3 was adjusted for age, sex, education, employment, Townsend Deprivation Index, drinking frequency, smoking status, body mass index, genetic risk, preexisting psychiatric disorders, cardiovascular diseases, and cancer. Mutual adjustment model was adjusted for Model 3 and 5 frailty components (mutual adjustment). HRs (95% CIs) and *P*-values were estimated using Cox proportional hazard models. Abbreviations: HR, hazards ratio; CI, confidence interval.

| Groups | Person-years | Cases/N | HRs (95% CIs) | | P values |
|---|---|---|---|---|---|
| **Low genetic risk** | | | | | |
| Non-frailty | 1,178,730 | 142/88,013 | 1(Reference) | | |
| Pre-frailty | 723,403 | 156/55,098 | 1.44 (1.15, 1.82) | | 0.002 |
| Frailty | 55,895 | 39/4,487 | 2.86 (1.99, 4.12) | | <0.001 |
| **Intermediate genetic risk** | | | | | |
| Non-frailty | 1,163,291 | 192/86,914 | 1.28 (1.03, 1.59) | | 0.026 |
| Pre-frailty | 730,210 | 260/55,624 | 2.18 (1.77, 2.68) | | <0.001 |
| Frailty | 63,325 | 54/5,111 | 2.93 (2.12, 4.07) | | <0.001 |
| **High genetic risk** | | | | | |
| Non-frailty | 1,112,557 | 236/83,212 | 1.56 (1.26, 1.92) | | <0.001 |
| Pre-frailty | 765,278 | 366/58,365 | 2.62 (2.15, 3.20) | | <0.001 |
| Frailty | 75,092 | 73/6,096 | 3.09 (2.30, 4.17) | | <0.001 |

HR (95% CI)

**Fig 2. Joint associations of frailty status and genetic risk with suicide attempt.** *Note*: Model adjusted for age, sex, education, employment, Townsend Deprivation Index, drinking frequency, smoking status, body mass index, preexisting psychiatric disorders, cardiovascular diseases, cancer, and the first 10 principal components of ancestry. HRs (95% CIs) and *P*-values were estimated using Cox proportional hazard models. Abbreviations: CI, confidence interval; HR, hazard ratio.

0.41%) to 3.53% (95% CI: 1.72%, 5.23%) (Fig 3). Collectively, these biomarkers explained 14.15% (95% CI: 8.94%, 19.74%; $P_{FDR}$ < 0.001) of the frailty-SA association. However, subsequent two-step MR analyses did not provide statistically significant evidence supporting a causal mediating role for these biomarkers (Table L in S2 Appendix).

## Sensitivity and subgroup analyses

Overall, the results of the sensitivity analyses were consistent with the primary findings, supporting the robustness of the observed association between frailty status and the risk of SA. Excluding SA cases that occurred within the first 2 years of follow-up did not materially change the results (Table M in S2 Appendix). Similarly, analyses based on multiply imputed datasets yielded effect estimates comparable to those from the complete-case analysis (Table N in S2 Appendix). When participants with baseline CVD, cancer, or psychiatric disorders and those with higher socioeconomic deprivation were excluded, the associations between frailty status and SA risk remained robust (Table O in S2 Appendix). Additional adjustment for the CCI also did not materially alter the results (Table P in S2 Appendix), suggesting that residual confounding by comorbidity burden was unlikely to fully explain the observed associations. *E*-value analyses indicated substantial robustness to unmeasured confounding, as an unmeasured confounder would need to be associated with both frailty status and SA by a risk ratio of ~3.0 to fully account for the observed associations (Table Q in S2 Appendix). Furthermore, results from the competing risk analysis accounting for death as a competing event were consistent with those from the primary Cox proportional hazards models (Table R in S2 Appendix). Stratified analyses by age, sex, education level, smoking status, drinking frequency, and BMI consistently showed persistent associations between frailty and SA risk, with no significant effect modification (all *P* for interaction > 0.05) (Table S in S2 Appendix). Scatter plots and LOO plots are presented in Fig E in S1 Appendix. Estimates from MR analyses using the MR-Egger, weighted median, random-effects IVW method, MR-Egger intercept test for horizontal pleiotropy, and heterogeneity tests with Cochran's Q statistic are provided in the Tables T and U in S2 Appendix.

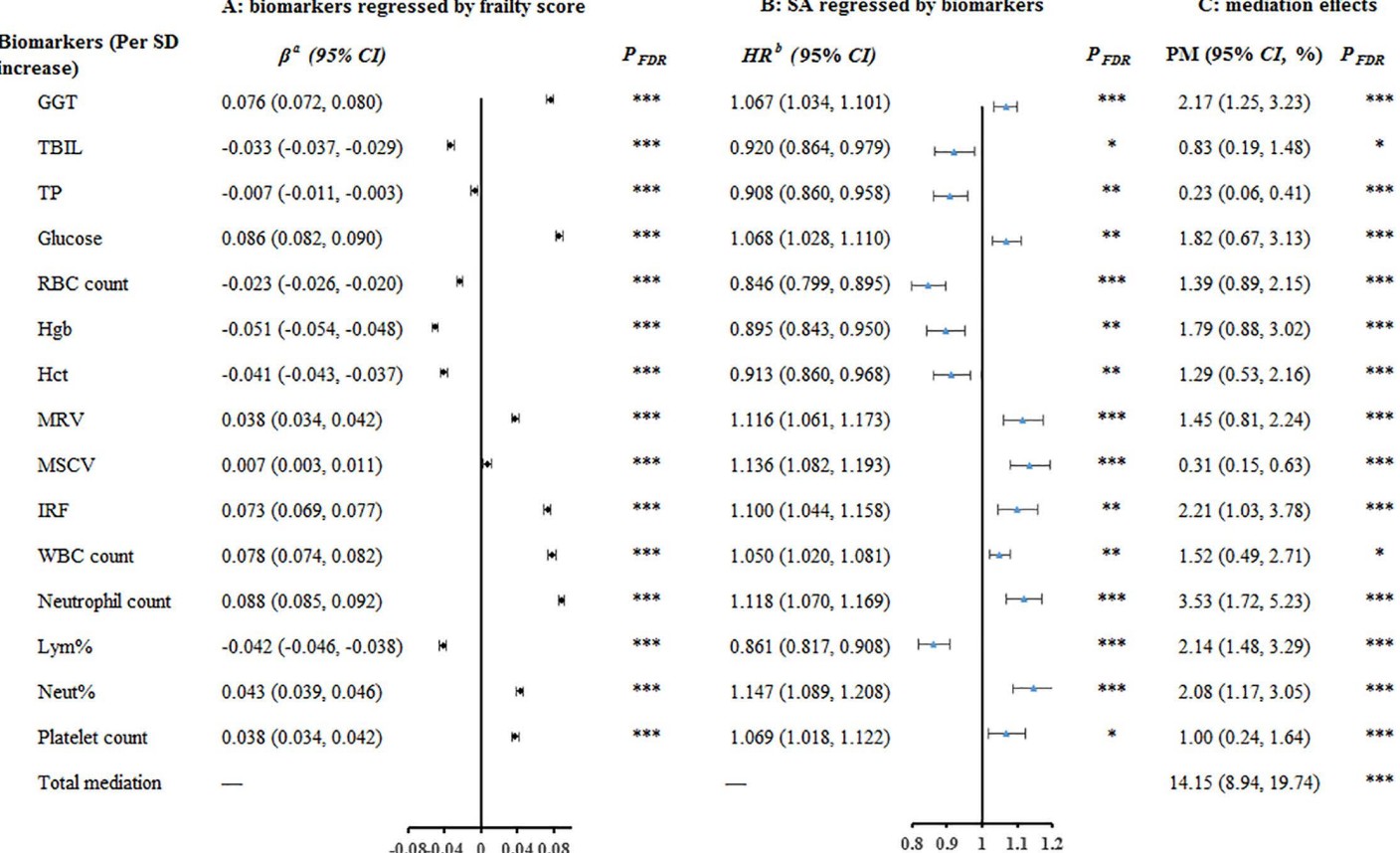

**Fig 3. Associations of frailty score with blood biomarkers(A), associations of blood biomarkers with suicide attempt (B), and mediating role of biomarkers in the association between frailty score and suicide attempt (C).** *Note*: [a]Linear regression models examined the associations of the frailty score with biomarkers, and adjusted for age, sex, education, employment, Townsend Deprivation Index, drinking frequency, smoking status, body mass index, genetic risk for SA, preexisting psychiatric disorders, cardiovascular diseases, and cancer; *P*-values for β coefficients were estimated from these linear regression models. [b]Cox proportional hazard models examined the associations between biomarkers and the risk of SA and adjusted for the above covariates plus the frailty score; *P*-values for HRs were estimated from these Cox models. The significance of the PM was estimated using a nonparametric bootstrap method with 1,000 resamples, and corresponding *P*-values were derived from this procedure. *FDR<0.05, **FDR<0.01, ***FDR<0.001. Abbreviations: SA, suicide attempt; SD, standard deviation; CI, confidence interval; HR, hazard ratio; PM, Proportion mediated; GGT, Gamma-glutamyltransferase; TBIL, Total bilirubin; TP, Total protein; RBC, Red blood cell; Hgb, Hemoglobin concentration; Hct, Hematocrit percentage; MRV, Mean reticulocyte volume; MSCV, Mean sphered cell volume; IRF, Immature reticulocyte fraction; WBC, White blood cell; Lym%, Lymphocyte percentage; Neut%, Neutrophil percentage.

## Discussion

Using data from the UK Biobank cohort, we found that physical frailty was associated with an increased risk of SA, independent of psychiatric disorders and genetic susceptibility. Two-sample MR analyses provided complementary genetic evidence supporting the observed association. Additionally, frailty and genetic risk interact additively to elevate SA risk, with the greatest relative risk observed in individuals with both frailty and high genetic risk. Furthermore, we found that ~14.2% of the frailty-SA association could be statistically explained by biomarkers related to liver function, metabolism, immune function, and inflammation, although MR did not support a causal mediating role.

This study confirmed the prospective association between FP-measured frailty and SA risk. Our findings complement and extend previous observations suggesting that frailty may be linked to psychiatric disorders or SI, both of which are

closely related to SA [7,17–19]. For example, a cross-sectional survey of 894 Chinese community-dwelling older adults identified a significant correlation between frailty and SI [18]. Another cross-sectional survey based on the National Health and Nutrition Examination Survey (NHANES) data observed a notable association of FI with self-reported SI [19]. In depressed patients, frailty has been shown to be associated with SI, independent of depression severity [17]. Additionally, MR analyses by Xiao and colleagues identified causal associations between frailty and several common mental disorders [7]. Notably, SA is more associated with suicide deaths than SI and psychiatric disorders, underscoring the critical need to identify potential risk factors for SA to facilitate early intervention and reduce mortality. Still, there is a significant lack of research investigating the association between frailty and SA. Although a recent review suggested that frailty may be an important underlying factor in SA during later life, the evidence is primarily based on qualitative studies [16], with only one quantitative study available [15]. Using follow-up data from a large-scale cohort of U.S. veterans aged 65 years and older, Kuffel and colleagues demonstrated that a cumulative FI, composed of dozens of deficits, was associated with a greater than 40% increased risk of SA, supporting our findings [15,20]. However, given the complexity and high cost of FI measurements, the user-friendly FP provides a more practical alternative for clinical and large-scale epidemiologic research. Despite its significance, there is still a notable gap in research investigating the relationship between FP and SA. The present study revealed that pre-frailty and frailty, as measured by FP, are significantly associated with an increased risk of developing SA. This association persisted even after adjusting for sociodemographic and lifestyle factors, psychiatric disorders, CVD, cancer, and genetic risk, as well as in sensitivity and subgroup analyses. Two-sample MR analyses provided additional support for the genetic association between FP-measured physical frailty and SA, suggesting that the observed association may be consistent with a potential causal relationship.

Frailty components, including exhaustion, weight loss, slow gait speed, and low grip strength, were independently associated with an elevated risk of SA. As expected, exhaustion, a key component of depression, exhibited the strongest association with SA, consistent with previous findings [21,46]. Following exhaustion, weight loss was tied to a 42% higher SA risk. Supporting this finding, a large cohort study with a mean follow-up of 6.9 years reported that weight loss was associated with an increased risk of suicide-related mortality in later life [23]. After adjusting for other frailty components, slow gait speed and low grip strength remained significantly associated with a 20% and 16% additional risk of SA, respectively. Similar to our results, prior cross-sectional studies have also documented associations between these factors and SI [17,22]. While previous studies have identified a significant association between low PA and SA [24,47], the current study did not observe this relationship. Specifically, low PA was significantly associated with an increased risk of SA when unadjusted for other FP components (HR = 1.27, 95% CI [1.11, 1.45]; P < 0.001). However, after further adjusting for other frailty components, the association was attenuated and lost statistical significance (HR = 1.13, 95% CI [0.98, 1.30]; P = 0.101). Since prior studies did not consider other frailty components [24,47], we speculated that the observed association between PA and SA may have been confounded by other factors. Overall, studies investigating the associations between frailty components and SA are quite limited and predominantly cross-sectional [17,21,22,24,46,47]. Our findings address this gap by providing novel prospective evidence on the association of specific frailty components with SA while accounting for other frailty components. These results may inform the development of tailored suicide prevention strategies for frail individuals.

The study applied the PRS for SA to examine the complex interplay between genetic risk, frailty status, and SA risk. Our findings revealed a synergistic interaction between frailty and genetic risk, leading to an increased risk of SA. Participants with both high genetic risk and frailty exhibited the highest SA risk, indicating that their combined effect exceeded the sum of their individual contributions. Furthermore, across genetic risk stratifications, nonfrail participants had a decreased SA risk than those with frailty. These findings highlight the potential benefits of improving frailty status in mitigating the risk of SA, especially among individuals with high genetic risk.

The precise mechanisms underlying the frailty-SA association remain unclear. While frailty is widely recognized as a risk factor for common mental disorders, CVD, and cancer [7,48], all of which are themselves associated with increased

SA risk [1,3,49]. Still, adjusting for these comorbidities only partially attenuated the observed association, suggesting that frailty may influence SA risk through additional pathways. One plausible explanation involves psychosocial mechanisms. Specifically, frailty is often accompanied by substantial psychological and social consequences, including depressive symptoms, anxiety, social isolation, hopelessness, and perceived burdensomeness [6,9,18]. These factors have been identified as proximal risk correlates of suicidal behavior [1,3]. These psychosocial sequelae may accumulate as frailty progresses, intensifying emotional distress, and diminishing perceived social connectedness and coping resources [50]. In line with this, we observed that frailty was also associated with SI, and that SI plays a key role in the frailty-SA relationship. Approximately one-third of the frailty-SA association could be explained by SI (Table V in S2 Appendix). Thus, in addition to physical vulnerability, frailty may contribute to SA through both psychosocial adversity and emotional distress that facilitate the transition from SI to SA. To further explore potential biological pathways, we investigated a series of circulating biomarkers and observed that ~14.2% of the frailty-SA association could be statistically attributed to biomarkers related to liver function, glucose metabolism, and blood cell and platelet levels. Notably, these alterations in biomarkers may not stem purely from biological factors. Psychological distress and chronic stress exposure, which are common among frail individuals [51], can influence neuroendocrine regulation and immune functioning [52,53], thereby shaping metabolic, inflammatory, and hematological profiles. As such, biomarker dysregulation may reflect an interplay between biological aging processes and stress-related psychobiological responses, both of which could contribute to elevated suicide risk. Among the identified potential mediating biomarkers, liver function biomarkers, including TP, GGT, and TBIL, showed modest but statistically significant indirect effects. These findings align with previous evidence linking frailty to impaired liver function and liver-related disorders [54], which have also been associated with SA risk [55]. For example, a large cohort study reported associations between frailty and a broad spectrum of liver diseases [54], while another study found that among depressed patients, those with liver disease had significantly higher lifetime SA rates than those without [55]. Together, these results support a potential pathway in which frailty-related hepatic dysfunction may contribute to vulnerability for SA. Additionally, biomarkers related to glucose metabolism and RBC indices contributed modestly to the frailty-SA association. Frailty has been linked to metabolic disorders (e.g., insulin resistance, hypoglycemia) and nutritional deficiencies (e.g., anemia), both of which increase SA risk [13,49,56,57]. Finally, immune and inflammatory biomarkers, including WBCs, neutrophils, and lymphocytes, partially mediated the frailty-SA association. Chronic inflammation and immune dysregulation, core biological features of frailty reflected in these biomarkers, may contribute to increased risk of SA [13,49,58]. Notably, subsequent two-step MR analyses did not provide statistically significant evidence supporting a causal mediating role for these biomarkers. The discrepancy between the observational mediation and MR findings may reflect several factors. First, genetic instruments for circulating biomarkers typically explain limited phenotypic variance, substantially reducing statistical power in MR analyses, particularly for rare and etiologically heterogeneous outcomes such as SA. Second, the biomarkers identified in observational analyses may primarily represent downstream correlates of frailty or shared biological processes, rather than direct causal mediators in the frailty-SA pathway. Therefore, these findings should be interpreted as indicating potential biological pathways rather than confirmed causal intermediates. Given that the identified biomarkers accounted for only a small proportion of the overall association, it is likely that other mechanisms, particularly psychosocial pathways such as emotional distress, reduced social support, and maladaptive stress coping, also play important roles. Future studies incorporating repeated measures of psychological symptoms, social factors, and biological markers are warranted to better disentangle these interrelated mechanisms, which may help identify modifiable targets for suicide prevention among frail individuals.

Several strengths of this study warrant mention. Given the lack of research examining the association between FP-measured frailty and SA risk, our study provides novel evidence on this matter. The reliability of our findings is strengthened by the prospective design, extended follow-up period, large cohort size, integration of prospective cohort and MR methodologies, comprehensive adjustment for key covariates (including genetic factors), and multiple additional analyses. Furthermore, mediation analyses were performed to provide deeper insights into the biological mechanisms

linking frailty to SA risk. Importantly, we incorporated the PRS for SA, demonstrating that alleviating frailty may confer a sustained benefit in reducing SA risk, even among individuals with high genetic susceptibility. However, the study has several limitations. First, despite prior evidence supporting the reproducibility of risk factor associations in the UK Biobank [59], the low response rate (5.5%) raises concerns about potential selection bias. Also, potential selection bias due to participant exclusion cannot be entirely ruled out. While $P$ values were statistically significant (all $P < 0.01$) for most comparisons due to the large sample size, the standardized mean differences were generally small, suggesting that the risk of substantial selection bias is likely limited (Table W in S2 Appendix). Second, although our analyses accounted for multiple confounders and followed participants for a median of 13.6 years, the possibility of unmeasured or residual confounders and reverse causality cannot be ruled out. Furthermore, observational research is not able to test the causality of associations. Nonetheless, our supplementary MR analyses leveraging genetic instruments provided complementary evidence that is consistent with the observed association, while helping to reduce the influence of residual confounding and reverse causality. Third, as the study population comprised only White participants, further research is required to validate these findings in diverse racial and ethnic groups. Additionally, the lack of validation in another independent dataset is a limitation. While the UK Biobank cohort is large and well-characterized, these findings should be replicated in other independent cohorts. Fourth, the reliance on self-reported data for most frailty components may introduce information bias, though previous studies have validated FP-measured frailty status [20,29]. Fifth, because the study outcome was determined based on hospital admission records, underreporting may have occurred due to unrecorded or undiagnosed cases. Finally, as both frailty and biomarkers were collected at baseline, establishing a causal temporal relationship through mediation analysis remains challenging. Future studies incorporating longitudinal repeated measures data could help confirm the mediating factors identified in this study.

In conclusion, the cohort study demonstrated that physical frailty was associated with an increased risk of SA, with MR analyses providing complementary genetic evidence consistent with this association. Besides, frailty and genetic risk interact additively to increase SA risk, with the greatest relative risk observed in frail individuals at high genetic risk. Furthermore, a range of biomarkers related to liver function, metabolism, immune function, and inflammation statistically explained a modest proportion of this association, highlighting potential biological pathways accompanying frailty. However, these biomarkers should be interpreted as potential contributors rather than confirmed causal mediators. Incorporating frailty assessment and management into primary suicide prevention strategies may facilitate the identification of high-risk individuals and inform future research into integrated biological and psychosocial interventions.

## Supporting information

**S1 Checklist. STROBE Statement—Checklist of items that should be included in reports of *cohort studies*.** The STROBE checklist is best used in conjunction with this article (freely available on the websites of PLoS Medicine at http://www.plosmedicine.org/, Annals of Internal Medicine at http://www.annals.org/, and Epidemiology at http://www.epidem.com/). Information on the STROBE Initiative is available at http://www.strobe-statement.org.
(DOCX)

**S2 Checklist. STROBE-MR checklist of recommended items to address in reports of *Mendelian randomization studies*.** This checklist is copyrighted by the Equator Network under the Creative Commons Attribution 3.0 Unported (CC BY 3.0) license.1Skrivankova VW, Richmond RC, Woolf BAR, Yarmolinsky J, Davies NM, Swanson SA, et al. Strengthening the Reporting of Observational Studies in Epidemiology using Mendelian Randomization (STROBE-MR) Statement. JAMA. 2021; under review. 2. Skrivankova VW, Richmond RC, Woolf BAR, Davies NM, Swanson SA, VanderWeele TJ, et al. Strengthening the Reporting of Observational Studies in Epidemiology using Mendelian Randomisation (STROBE-MR): Explanation and Elaboration. BMJ. 2021;375:n2233.
(DOCX)

**S1 Text. Supplementary methods.**
(DOCX)

**S1 Appendix. Figures A–E. Fig A.** Flowchart of the selection of the study population from the UK Biobank study. **Fig B.** Directed acyclic graph depicting the hypothesized causal structure underlying the association between physical frailty and suicide attempt. Sociodemographic factors included age, sex, educational level, employment status, and Townsend Deprivation Index; lifestyle factors included smoking status, drinking frequency, and body mass index; medical histories included cardiovascular diseases, psychiatric disorders, and cancer. **Fig C.** Crude cumulative incidence of suicide attempt by frailty status. SA, suicide attempt. The *P* value was estimated using the log-rank test. **Fig D.** Dose-response associations between frailty scores and the risk of suicide attempt. SA, suicide attempt; CI, confidence interval; HR, hazard ratio. Model was adjusted for age, sex, education, employment, Townsend Deprivation Index, drinking frequency, smoking status, body mass index, genetic risk, preexisting psychiatric disorders, cardiovascular diseases, and cancer. *P*-values for overall and nonlinear associations were estimated using restricted cubic spline analysis based on the Cox proportional hazards model. **Fig E.** Scatter plot and leave-one-out test for the causal association between physical frailty and suicide attempt. MR, Mendelian randomization; IVW, inverse-variance weighted; SNP, single nucleotide polymorphism. Causal effect estimates in the scatter plot were obtained using the IVW method, IVW with multiplicative random-effects, MR-Egger regression, and the weighted median method. Corresponding statistical significance was assessed using these respective methods. The leave-one-out analysis was conducted using the IVW method, with each SNP sequentially removed. Horizontal lines represent 95% confidence intervals.
(DOCX)

**S2 Appendix. Tables A–W. Table A.** Frailty definition and cutoff values in the UK Biobank. **Table B.** Detailed information on selected blood biomarkers in the study. **Table C.** Measures of covariates at baseline in the UK Biobank. **Table D.** Summary of missing data for covariates. **Table E.** Detailed information on selected genetic variants and their associations with the exposure and outcome. **Table F.** List of the 17 comorbidities, assigned weights and associated ICD-10 codes for construction of the Charlson Comorbidity Index. **Table G.** Baseline characteristics according to suicide attempt status. **Table H.** Risk of suicide attempt according to frailty status within genetic risk categories. **Table I.** RERI and AP for additive interaction between frailty status and genetic risk. **Table J.** Multivariable-adjusted Cox regression models for the association of blood biomarkers with suicide attempt. **Table K.** Multivariable-adjusted linear regression models for the associations between frailty scores and suicide attempt-related blood biomarkers. **Table L.** Two-step Mendelian randomization analyses evaluating the potential causal roles of candidate biomarkers in the association between physical frailty and suicide attempt. **Table M.** Associations of frailty status with the risk of suicide attempt after excluding the cases that occurred within the first 2-years of follow-up. **Table N.** Associations of frailty status with the risk of suicide attempt after using multiple imputation for missing covariates. **Table O.** Associations of frailty status with the risk of suicide attempt after excluding participants with baseline CVD, cancer, or higher deprivation. **Table P.** Associations of frailty status with the risk of suicide attempt after further adjustment for the Charlson Comorbidity Index. **Table Q.** E-values for the associations between frailty status and suicide attempt. **Table R.** Associations of frailty status with the risk of suicide attempt accounting for death as a competing risk. **Table S.** Stratified analysis for the association of the frailty status with the risk of suicide attempt. **Table T.** Estimates from MR analysis using the IVW method for the association between frailty and suicide attempt, and replicated estimates from MR analyses using the MR-Egger regression and weighted median methods for the same association. **Table U.** Replicated estimates from MR analysis using the random-effects IVW method for the association between frailty and suicide attempt, along with MR-Egger intercept test results for horizontal pleiotropy and heterogeneity test results using Cochran's Q statistic. **Table V.** Associations between physical frailty and suicidal ideation, and mediating role of suicidal ideation in the frailty-SA association. **Table W.** Comparison of baseline characteristics between included and excluded populations.
(DOCX)

## Acknowledgments

We thank all participants and staff from the UK Biobank study.

## Author contributions

**Conceptualization:** Bao-Peng Liu, Cun-Xian Jia.

**Data curation:** Bao-Peng Liu.

**Formal analysis:** Wei Hu, Li-jie Gao.

**Funding acquisition:** Bao-Peng Liu, Cun-Xian Jia.

**Methodology:** Wei Hu, Tian-Shu Liu, Bao-Peng Liu, Cun-Xian Jia.

**Project administration:** Bao-Peng Liu, Cun-Xian Jia.

**Software:** Wei Hu, Tian-Shu Liu.

**Supervision:** Bao-Peng Liu, Cun-Xian Jia.

**Validation:** Li-jie Gao, Ge Tian, Jia-Ning Wang, Yu-Bin Ma, Zi-Ang Zheng, Tong-Jie Feng, Xiao-Xin Niu, Bao-Peng Liu.

**Visualization:** Wei Hu, Li-jie Gao, Tian-Shu Liu.

**Writing – original draft:** Wei Hu.

**Writing – review & editing:** Li-jie Gao, Tian-Shu Liu, Ge Tian, Jia-Ning Wang, Yu-Bin Ma, Zi-Ang Zheng, Tong-Jie Feng, Xiao-Xin Niu, Yi-Ning Yan, Bao-Peng Liu, Cun-Xian Jia.

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
