## [Editor Report · Decision Letter 0]

23 Sep 2025

Dear Dr Hu,

Thank you for submitting your manuscript entitled "Physical frailty, genetic risk, mediating biomarkers, and risk of suicide attempts: a prospective cohort study" for consideration by PLOS Medicine.

Your manuscript has now been evaluated by the PLOS Medicine editorial staff as well as by an academic editor with relevant expertise and I am writing to let you know that we would like to send your submission out for external peer review.

For clinical studies, please upload a copy of your trial study protocol as a supporting information file. The study protocol should be the version submitted for approval to the institutional review board or ethics committee, should include any amendments to the study protocol, as well as the date of their approval by the institutional review or ethics committee. Please also detail any deviations from the study protocol in the Methods section of your manuscript. The editors will consider the protocol and study conduct prior to a final decision for external review.

Please re-submit your manuscript within two working days, i.e. by Sep 25 2025 11:59PM.

Kind regards,

Jennifer Thorley

PLOS Medicine

---

## [Decision Letter · Decision Letter 1]

13 Jan 2026

Dear Dr Hu,

Many thanks for submitting your manuscript "Physical frailty, genetic risk, mediating biomarkers, and risk of suicide attempts: a prospective cohort study" (PMEDICINE-D-25-03230R1) to PLOS Medicine. The paper has been reviewed by subject experts and a statistician; their comments are included below and can also be accessed here: [LINK]

As you will see, the reviewers have provided their feedback, with minor and some more critical concerns about the methodology and the work described here. In summary, further clarifications and explanations have been requested in terms of the methodology and the introduction to the main question, and some more critical points were raised about frailty being a potential mediator and not the main driver of the association investigated here. After discussing the paper with the editorial team and an academic editor with relevant expertise, I'm pleased to invite you to revise the paper in response to the reviewers' comments. You will find the detailed reviews, as well as comments from the editorial team and the academic editor at the end of the letter. We plan to send the revised paper to some or all of the original reviewers, and we cannot provide any guarantees at this stage regarding publication.

We ask that you submit your revision by Feb 03 2026 11:59PM. However, if this deadline is not feasible, please contact me by email, and we can discuss a suitable alternative.

Don't hesitate to contact me directly with any questions (efourli@plos.org).

Best regards,

Evangelia

Evangelia Fourli,

Associate Editor

PLOS Medicine

efourli@plos.org

Comments from the academic editor and the editorial team:

Please acknowledge the lack of validation in an independent dataset as a limitation of the study.

Comments from the reviewers:

Reviewer #1: Review of the article PMEDICINE-D-25-03230R1 "Physical frailty, genetic risk, mediating biomarkers, and risk of suicide attempts: a prospective cohort study"

This article is a prospective analysis of the association of frailty and suicidal attempts in the UK Biobank cohort study. The authors also analysed the modifying and mediating roles of genetic risk and biomarkers. They found that pre-frail and frail status was associated with increased risk of suicidal attempts.

The article is well-written, the methodology is robust, the research question is relevant and quite novel, and the results are derived from the data.

Some comments:

1. There is a missing reference:

Draper B, Wand APF. Suicide in later life: the role of frailty and depression. Curr Opin Psychiatry. 2025;38(5):383-388. doi:10.1097/YCO.0000000000001009

2. In the sensitivity analyses, the authors used multiple imputation to deal with missing covariates. Can they give more details about the imputation procedure? Did they apply Rubin's rules to pool estimates and calculate confidence intervals? What was the percentage of missing values for each covariate?

3. How were the variables for adjustment chosen? Was a directed acyclic graph (DAG) used for the identification of confounders, mediators and colliders?

4. How were the selected biomarkers selected?

Reviewer #2: This is an interesting study on the association among physical frailty, genetic risk, mediating biomarkers, and risk of suicide attempt. However, there are a few major issues needing attention.

1. Not sure or convinced by the premise of the study that physical frailty is the main driver of suicide attempt with all the other factors as either covariates or mediation factors. It could be other way around that people with severe long-term conditions both physically and/or mentally plus siocioeconomic disadvantages are more likely to develop frailty as both a condition and a set of symptoms in the course leading to SA, therefore frailty could be a mediator itself for assessing the associations of the above mentioned diseases.

2. Table 2 - Associations of frailty status with the risk of suicide attempts, is the main results table of the study. However, only Model 3 is needed and rest models are redundant. Two issues, 1) Without fully adjusting for all the co-morbidities with detailed subtype, serverity and stages, the reliability and robustness of results of Model 3 are subject to scrutiny. So far only pre-existing psychiatric disorders, cardiovascular diseases, and cancer were included in the model, and it is far from comprehensive and complete.

3. A complete results table for Table 2 is needed so that we can see the contribution from all the factors, especially the PRS contribution is missing from the table.

4. Competing risk. How many people died during the 13-year follow up? For survival analysis with an end point other that all cause mortality, a competing risk analysis is needed (competing risk due to death).

5. Page 5, it says "participants with incomplete information on frailty components (n = 14,824) were excluded". This could bias the results. A comparison table is needed to compare those included with those excluded

6. MR analysis is difficult to follow, and a bit redundant as it seems to force the causality issue between fraity and SA, which doesn't really make sense.

Reviewer #3: Hu and colleagues investigate the role of frailty in the risk of suicide attempt. This is an interesting goal, and the approach used is justified. However, there are several issues that the authors should consider to increase the impact of their investigation.

1. In addition to suicide attempt, it could be interesting to assess whether frailty has a similar impact on suicidal ideation and whether the effect on suicidal attempt is related at least partially to its relationship with suicidal ideation. This may have important public health implications.

2. The biomarker analysis is interesting. However, the authors should consider expanding it further using a one-sample MR design to assess whether the associations observed are due to cause-and-effect relationships.

3. In addition to the STROBE checklist for genetic studies, the authors should report whether their MR analyses are in line with the STROBE-MR checklist (PMID: 34698778).

4. The authors provide limited information regarding the genetically informed ancestry assignment to define their cohort. Additionally, they use incorrect terminology to describe population groups (e.g., "... constructed for White populations"). They should follow the recommendations of the National Academies regarding population descriptors for genetic studies (PMID: 36989389). Also, I recommend them the genetically informed ancestry assignment of the Pan-UKBB Initiative (PMID: 40968291).

5. Since the suicide-attempt GWAS of the Psychiatric Genomics Consortium includes the UK Biobank, the authors should clearly state that they obtained a version of the GWAS data that exclude UK Biobank, and this is different from the one described in the reference cited.

6. Hazard model analysis was conducted using SA information available from electronic health records. However, in the methods, the authors mentioned that self-reported information was also used to define SA cases at baseline. The authors should clarify this point to clearly explain how these different data sources were used.

7. In the main text, the authors should expand the description of the sensitivity analyses in the main text to guide the readers through the supplemental material describing them.

Reviewer #4: This study examined the association between frailty (measured using an index composed of a series of indicators) and suicide attempt in the UK Biobank. More specifically, it merged UK Biobank survey data with administrative data on suicide attempts prospectively collected over a median follow-up of 13 years. The study further analyzed potential biological mediation mechanisms using blood biomarkers. Mendelian randomization was used to strengthen causal inference regarding the association between frailty and suicide attempt. Overall, the paper is well written and methodologically sound, and, in my opinion, it represents an important addition to the evidence on risk factors for suicide attempt. The authors appropriately identify methodological limitations.

My main suggestions are:

* The Introduction could be reworked to provide more substantive justification for the analyses conducted in this study. In several instances, the rationale relies on the fact that a topic "has not been investigated before," which, on its own, is not a sufficiently strong justification.

* Using multivariable Mendelian randomization to test the mediating role of the identified biomarker could be a valuable addition to the paper and would help address the limitation related to the simultaneous measurement of frailty and biomarkers.

* Frailty can be associated with important psychological consequences, including depressive symptoms, anxiety, social isolation, and feelings of hopelessness or perceived burdensomeness, which may influence individuals' risk of suicide attempt. These factors deserve to be mentioned and discussed as alternative or complementary mechanisms underlying the associations observed in this study. Changes in biomarkers may also be related to such psychological states.

Other points:

* Page 3: "SA, one of the strongest risk factor for suicide deaths, is preventable." Suggest rephrasing as "may be preventable."

* Same page: "Suicide attempt increase with age… risk factor for SA." The logic of this sentence is unclear and may benefit from rephrasing for clarity.

---

* Please upload any figures associated with your paper as individual TIF or EPS files with 300dpi resolution at resubmission; please read our figure guidelines for more information on our requirements: http://journals.plos.org/plosmedicine/s/figures. While revising your submission, we strongly recommend that you use PLOS's NAAS tool (https://ngplosjournals.pagemajik.ai/artanalysis) to test your figure files. NAAS can convert your figure files to the TIFF file type and meet basic requirements (such as print size, resolution), or provide you with a report on issues that do not meet our requirements and that NAAS cannot fix.

After uploading your figures to PLOS's NAAS tool - https://ngplosjournals.pagemajik.ai/artanalysis, NAAS will process the files provided and display the results in the "Uploaded Files" section of the page as the processing is complete.

If the uploaded figures meet our requirements (or NAAS is able to fix the files to meet our requirements), the figure will be marked as "fixed" above. If NAAS is unable to fix the files, a red "failed" label will appear above.

When NAAS has confirmed that the figure files meet our requirements, please download the file via the download option, and include these NAAS processed figure files when submitting your revised manuscript.

* Please ensure that the study is reported according to the STROBE guideline and include the completed STROBE checklist as Supporting Information. When completing the checklist, please use section and paragraph numbers, rather than page numbers. Please add the following statement, or similar, to the Methods: "This study is reported as per STROBE guideline (S1 Checklist)."

FIGURES AND TABLES

SUPPLEMENTARY MATERIAL

REFERENCES

OBSERVATIONAL STUDIES

* Abstract: Please include the study design, population and setting, number of participants, years during which the study took place (enrollment and follow up), length of follow up, and main outcome measures.

* Please ensure that the study is reported according to the STROBE (or appropriate STOBE extension) guideline (available from: https://www.equator-network.org/reporting-guidelines/strobe) and include the completed STROBE (or STROBE extension) checklist as Supporting Information. Please add the following statement, or similar, to the Methods: "This study is reported as per the Strengthening the Reporting of Observational Studies in Epidemiology (STROBE) guideline (S1 Checklist)." When completing the checklist, please use section and paragraph numbers, rather than page numbers.

* [FOR POPULATION HEALTH/REGISTRY STUDIES] Please ensure that the study is reported according to the RECORD guideline (available from https://www.record-statement.org) and include the completed checklist as Supporting Information. Please add the following statement, or similar, to the Methods: "This study is reported as per the Reporting of Studies Conducted using Observational Routinely-Collected Data (RECORD) guideline (S1 Checklist)." When completing the checklist, please use section and paragraph numbers, rather than page numbers.

* [FOR POPULATION HEALTH ESTIMATES] Please ensure that the study is reported according to the GATHER statement (available from https://www.equator-network.org/reporting-guidelines/gather-statement) and include the completed checklist as Supporting Information. Please add the following statement, or similar, to the Methods: "This study is reported as per the Guidelines for Accurate and Transparent Health Estimates Reporting (GATHER) statement (S1 Checklist)." When completing the checklist, please use section and paragraph numbers, rather than page numbers.

* [FOR MEDIATION ANALYSES] We recommend that the study is reported according to the AGReMA statement (https://agrema-statement.org/#:~:text=AGReMA%20is%20an%20evidence%2D%20and,randomised%20trials%20and%20observational%20studies) and include the completed checklist as Supporting Information. Please add the following statement, or similar, to the Methods: "This study is reported as per the Guideline for Reporting Mediation Analyses (AGReMA) statement (S1 Checklist)." When completing the checklist, please use section and paragraph numbers, rather than page numbers.

* For all observational studies, in the manuscript text, please indicate: (1) the specific hypotheses you intended to test, (2) the analytical methods by which you planned to test them, (3) the analyses you actually performed, and (4) when reported analyses differ from those that were planned, transparent explanations for differences that affect the reliability of the study's results. If a reported analysis was performed based on an interesting but unanticipated pattern in the data, please be clear that the analysis was data driven.

* Please state in the Methods section whether the study had a prospective protocol or analysis plan. If a prospective analysis plan (from your funding proposal, IRB or other ethics committee submission, study protocol, or other planning document written before analyzing the data) was used in designing the study, please include the relevant document(s) with your revised manuscript as a Supporting Information file to be published alongside your study and cite it in the Methods section. A legend for this file should be included at the end of your manuscript. If no such document exists, please make sure that the Methods section transparently describes when analyses were planned, and when/why any data-driven changes to analyses took place. Changes in the analysis, including those made in response to peer review comments, should be identified as such in the Methods section of the paper, with rationale.

MENDELIAN RANDOMIZATION STUDIES

* Please ensure that the study is reported according to the STROBE-MR guideline (https://www.equator-network.org/reporting-guidelines/strobe/) and include the completed STROBE-MR checklist as Supporting Information. Please add the following statement, or similar, to the Methods: "This study is reported as per the Strengthening the Reporting of Observational Studies in Epidemiology (STROBE) guideline, specific for mendelian randomization (S1 Checklist)." When completing the checklist, please use section and paragraph numbers, rather than page numbers.

* In the Introduction, please describe the exposure and the evidence for a potential causal relationship between exposure and outcome.

* In the Methods, please explicitly state the 3 core instrumental variable assumptions for the main analysis (relevance, independence, and exclusion restriction), as well assumptions for any additional or sensitivity analysis.

* In the Methods, please describe the MR estimator (e.g., 2-stage least squares, Wald ratio) and related statistics. Detail the included covariates and, in case of 2-sample MR, whether the same covariate set was used for adjustment in the 2 samples.

* If you are presenting an instrumental variable estimate, please compare this to the conventional observational estimate.

* Report the associations between genetic variant and exposure and between genetic variant and outcome, preferably on an interpretable scale.

* Report MR estimates of the relationship between exposure and outcome and the measures of uncertainty from the MR analysis, on an interpretable scale, such as odds ratio or relative risk per SD difference.

* If relevant, please consider translating estimates of relative risk into absolute risk for a meaningful time period.

* Please consider including plots to visualize results (e.g., forest plot, scatterplot of associations between genetic variants and outcome vs between genetic variants and exposure).

---

## [Decision Letter · Decision Letter 2]

11 Mar 2026

Dear Dr. Hu,

Thank you very much for re-submitting your manuscript "Physical frailty, genetic risk, mediating biomarkers, and risk of suicide attempt: a prospective cohort study" (PMEDICINE-D-25-03230R2) for review by PLOS Medicine.

I have discussed the paper with my colleagues and the academic editor and it was also seen again by xxx reviewers. I am pleased to say that provided the remaining editorial and production issues are dealt with we are planning to accept the paper for publication in the journal.

[LINK]

We look forward to receiving the revised manuscript by Mar 18 2026 11:59PM.

Sincerely,

Evangelia Fourli, Ph.D.

Senior Editor

PLOS Medicine

plosmedicine.org

Requests from Editors:

Please notice that some of the following requests may not apply to your work:

- Please ensure that any additional analysis in response to reviewer's comments included in the rebuttal is also included in the main or supplemental information of the manuscript (if not done already).

- Please include URLs for each funder

EDITORIAL REQUESTS

"* At this stage, we ask that you include a short, non-technical Author Summary of your research to make findings accessible to a wide audience that includes both scientists and non-scientists. The Author Summary should immediately follow the Abstract in your revised manuscript. This text is subject to editorial change and should be distinct from the scientific abstract. Ideally each sub-heading should contain 2-3 single sentence, concise bullet points containing the most salient points from your study. In the final bullet point of ‘What Do These Findings Mean?’ Please include the main limitations of the study in non-technical language.

Please see our author guidelines for more information: https://journals.plos.org/plosmedicine/s/revising-your-manuscript#loc-author-summary."

* Please confirm that your title complies with PLOS Medicine's style. Your title must be nondeclarative and not a question. It should begin with main concept if possible. "Effect of" should be used only if causality can be inferred, i.e., for an RCT. Please place the study design ("A randomized controlled trial," "A retrospective study," "A modelling study," etc.) in the subtitle (ie, after a colon).

* Please confirm that your abstract complies with our requirements, including format (three sections: Background, Methods and Findings, and Conclusions) and providing all the information relevant to this study type https://journals.plos.org/plosmedicine/s/submission-guidelines#loc-abstract

* Please ensure that the Introduction ends with a clear description of the study question or hypothesis.

* Please ensure that all abbreviations are defined at first use throughout the text.

* Please confirm that all numbers presented in the abstract are present and identical to numbers presented in the main manuscript text.

GENERAL

* Please review your text for claims of novelty or primacy (e.g. 'for the first time') and remove this language. In addition, please check that any use of statistical terms (such as trend or significant) are supported by the data, and if not please remove them.

* Please remove the 'conclusions' subheading from the discussion. Please also remove any other subheadings from the discussion.

"* Statistical reporting: Please revise throughout the manuscript, including tables and figures.

- Please report statistical information as follows to improve clarity for the reader ""22% (95% CI [13,28]; p</=)"".

- Please separate upper and lower bounds with commas instead of hyphens as the latter can be confused with reporting of negative values.

- Please repeat statistical definitions (HR, CI etc.) for each set of parentheses."

* In the abstract, please include the important dependent variables that are adjusted for in the analyses.

* In the author summary, in the final bullet point of 'What Do These Findings Mean?', please include the main limitations of the study in non-technical language.

* Please include an Acknowledgments section in your manuscript.

FUNDING STATEMENT

* The funding statement should include: specific grant numbers, initials of authors who received each award, URLs to sponsors’ websites. Also, please state whether any sponsors or funders (other than the named authors) played any role in study design, data collection and analysis, the decision to publish, or preparation of the manuscript. If they had no role in the research, include this sentence: “The funders had no role in study design, data collection and analysis, decision to publish, or preparation of the manuscript.”

COMPETING INTERESTS STATEMENT

* All authors must declare their relevant competing interests per the PLOS policy, which can be seen here: https://journals.plos.org/plosmedicine/s/competing-interests For authors with ties to industry, please indicate whether any of the interests has a financial stake in the results of the current study.

ETHICS AND CONSENT

* Please specify whether informed consent was written or oral. Please ensure that the research complies with the PLOS policy in full: https://journals.plos.org/plosmedicine/s/human-subjects-research#loc-patient-privacy-and-informed-consent-for-publication

FIGURES AND TABLES

* Please show graph axes beginning at zero. If this is not possible, please show a break in the axis.

* When a p value is given, please specify the statistical test used to determine it in the legend.

* Please provide the unadjusted comparisons as well as the adjusted comparisons in all relevant Tables

* Please specify the variables controlled for in all relevant Tables

OBSERVATIONAL, COHORT, CROSS-SECTIONAL, AND CASE CONTROL STUDIES

"* Did your study have a prospective protocol or analysis plan? Please state this (either way) early in the Methods section.

c) In either case, changes in the analysis-- including those made in response to peer review comments-- should be identified as such in the Methods section of the paper, with rationale."

Comments from Reviewers:

Reviewer #2: Thanks authors for their effort to address my questions. However, overall the responses are difficult to follow and not satisfactory to me. The reponses on co-morbidity, e-value, competing risk, missing values and MR are all patchy and not sufficiently convincing, and many response tables seemed only for the reviewers to see which is inadequate, and many sensitivity analyses should be the main analyses instead, eg. model 3 with co-morbidity index (but exclude those limited conditions), and competing risk model. The authors mostly argue on every single point and prove they have done the right thing, but it is unneccessary. Seriously and carefully acknowledging the limitations is also part of scentific research and will be appreciated.

Reviewer #3: The authors adequately addressed my previous comments.

Reviewer #4: Thank you for replying to my comment.

I think that paper has improved, and I have no further comments

[LINK]

---

## [Editor Report · Decision Letter 3]

26 Mar 2026

Dear Dr Hu,

On behalf of my colleagues and the Academic Editor, Dr Balbuena, I am pleased to inform you that we have agreed to publish your manuscript "Physical frailty, genetic risk, mediating biomarkers, and risk of suicide attempt: a prospective cohort study" (PMEDICINE-D-25-03230R3) in PLOS Medicine.

PRESS

Sincerely,

Evangelia Fourli, Ph.D.

Associate Editor

PLOS Medicine